# DualNet: Continual Learning, Fast and Slow

**Quang Pham**[1], **Chenghao Liu**[2], **Steven C.H. Hoi** [1,2]

[1] Singapore Management University
hqpham.2017@smu.edu.sg
[2] Salesforce Research Asia
{chenghao.liu, shoi}@salesforce.com

## Abstract

According to Complementary Learning Systems (CLS) theory [37] in neuroscience, humans do effective *continual learning* through two complementary systems: a fast learning system centered on the hippocampus for rapid learning of the specifics and individual experiences, and a slow learning system located in the neocortex for the gradual acquisition of structured knowledge about the environment. Motivated by this theory, we propose a novel continual learning framework named "DualNet", which comprises a fast learning system for supervised learning of pattern-separated representation from specific tasks and a slow learning system for unsupervised representation learning of task-agnostic general representation via a Self-Supervised Learning (SSL) technique. The two fast and slow learning systems are complementary and work seamlessly in a holistic continual learning framework. Our extensive experiments on two challenging continual learning benchmarks of CORE50 and miniImageNet show that DualNet outperforms state-of-the-art continual learning methods by a large margin. We further conduct ablation studies of different SSL objectives to validate DualNet's efficacy, robustness, and scalability. Code is publicly available at https://github.com/phquang/DualNet.

## 1 Introduction

Humans have the remarkable ability to learn and accumulate knowledge over their lifetime to perform different cognitive tasks. Interestingly, such a capability is attributed to the complex interactions among different interconnected brain regions [14]. One prominent model is the *Complementary Learning Systems (CLS) theory* [37, 30] which suggests the brain can achieve such behaviors via two learning systems of the "hippocampus" and the "neocortex." Particularly, the hippocampus focuses on fast learning of pattern-separated representation of specific experiences. Via the memory consolidation process, the hippocampus's memories are transferred to the neocortex over time to form a more general representation that supports long-term retention and generalization to new experiences. The two fast and slow learning systems always interact to facilitate fast learning and long-term remembering. Although deep neural networks have achieved impressive results [31], they often require having access to a large amount of i.i.d data while performing poorly on the *continual learning* scenarios over streams of task [19, 29, 36]. Therefore, the main focus of this study is exploring how the CLS theory can motivate a general continual learning framework with a better trade-off between alleviating catastrophic forgetting and facilitating knowledge transfer.

In literature, several continual learning strategies are inspired from the CLS theory principles, from using the episodic memory [36] to improving the representation [26, 44]. However, such techniques mostly use a single backbone to model both the the hippocampus and neocortex, which binds two representation types into the same network. Moreover, such networks are trained to minimize the supervised loss, they lack a separate and specific slow learning component that supports general representation learning. During continual learning, the representation obtained by repeatedly

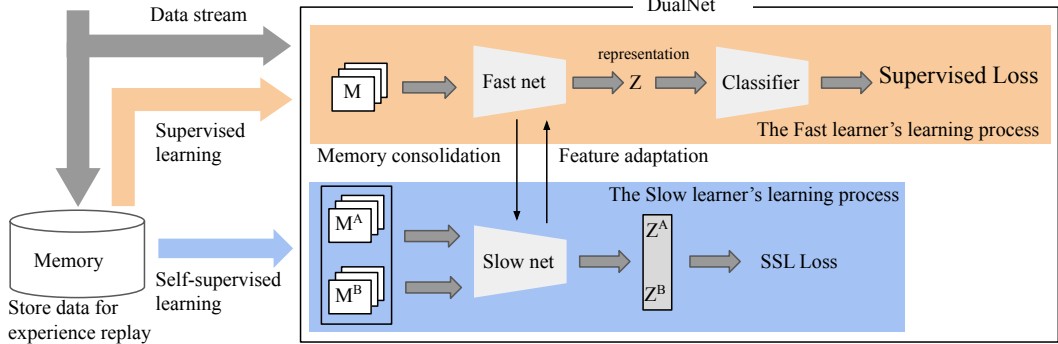

Figure 1: Overview of the DualNet' architecture. DualNet consists of (i) a slow learner (in blue) that learns representation by optimizing an SSL loss using samples from the memory, and (ii) a fast learner (in orange) that adapts the slow net's representation for quick knowledge acquisition of labeled data. Both learners can be trained synchronously.

performing supervised learning on a small amount of memory data can be prone to overfitting and may not generalize well across tasks. Consider that in continual learning, unsupervised representation [20, 40] is often more resisting to forgetting compared to supervised representation, which yields little improvements [25]; we propose to decouple representation learning from the supervised learning into two separate systems. To achieve this goal, in analogy to the slow learning system in neocortex, we propose to implement the slow general representation learning system using Self-Supervised Learning (SSL) [39]. Note that recent SSL works focus on the pre-training phase, which is not trivial to apply for continual learning as it is extensive in both storage and computational cost [28]. We argue that SSL should be incorporated into the continual learning process while decoupling from the supervised learning phase into two separate systems. Consequently, the SSL's slow representation is more general and can capture the intrinsic characteristics from data, which facilitates better generalization to both old and new tasks.

Inspired by the CLS theory [37], we propose *DualNet* (for Dual Networks, depicted in Figure 1), a novel framework for continual learning comprising *two* complementary learning systems: a *slow learner* that learns generic features via self-supervised representation learning, and a *fast learner* that adapts the slow learner's features to quickly attain knowledge from labeled samples via a novel per-sample based adaptation mechanism. During the supervised learning phase, an incoming labeled sample triggers the fast learner to make predictions by querying and adapting the slow learner's representation. Then, the incurred loss will be backpropagated through *both learners* to consolidate the current supervised learning pattern for long-term retention. Concurrently, the slow learner is always trained in the background by minimizing an SSL objective using only the memory data. Therefore, the slow and fast networks learning are completely *synchronous*, allowing DualNet to continue to improve its representation power even in practical scenarios where labeled data are delayed [13] or even limited, which we will demonstrate in Section 4.6. Lastly, we focus on developing DualNet for the online continual learning settings [36, 2] since it is more challenging to optimize deep networks in such scenarios [51, 3]. In the batch continual learning setting [46], the model is allowed to revisit data within the current task and can achieve good representations when learning the current task.

In summary, our work makes the following contributions:

1. We propose DualNet, a novel continual learning framework comprising two key components of fast and slow learning systems, which closely models the CLS theory.

2. We propose a novel learning paradigm for DualNet to efficiently decouple the representation learning from supervised learning. Specifically, the slow learner is trained in the background with SSL to maintain a general representation. Concurrently, the fast learner is equipped with a novel adaptation mechanism to quickly capture new knowledge. Notably, unlike existing adaptation techniques, our proposed mechanism does not require the task identifiers.

3. We conduct extensive experiments to demonstrate DualNet's efficacy, robustness to the slow learner's objectives, and scalability to the computational resources.

## 2 Method

### 2.1 Setting and Notations

We consider the online continual learning setting [36, 8] over a continuum of data $\mathcal{D} = \{\boldsymbol{x}_i, t_i, y_i\}_i$, where each instance is a labeled sample $\{\boldsymbol{x}_i, y_i\}$ with an *optional* task identifier $t_i$. Each labeled data sample is drawn from an underlying distribution $P^t(\boldsymbol{X}, \boldsymbol{Y})$ that represents a task and can suddenly change to $P^{t+1}$, indicating a task switch. When the task identifier $t$ is given as an input, the setting follows the *task-aware setting* where only the corresponding classifier is selected to make a prediction [36]. When the task identifier is not provided, the model has a shared classifier for all classes observed so far, which follows the *task-free setting* [7, 2]. We consider both scenarios in our experiments. A common continual learning strategy is employing an episodic memory $\mathcal{M}$ to store a subset of observed data and interleave them when learning the current samples [36, 9]. From $\mathcal{M}$, we use $\boldsymbol{M}$ to denote a randomly sampled mini-batch, and $\boldsymbol{M}^A$, $\boldsymbol{M}^B$ to denote two views of $\boldsymbol{M}$ obtained by applying two different data transformations. Lastly, we denote $\phi$ as the parameter of the slow network that learns general representation from the input data and $\boldsymbol{\theta}$ as the parameter of the fast network that learns the transformation coefficients.

### 2.2 DualNet Architecture

DualNet learns the data representation independent of the task's label, which allows for better generalization capabilities across tasks in the continual learning scenario. The model consists two main learning modules (Figure 1): (i) the slow learner is responsible for learning a general, task-agnostic representation; and (ii) the fast learner learns with labeled data from the continuum to quickly capture the new information and then consolidate the knowledge to the slow learner.

DualNet learning can be broken down into two *synchronous* phases. First, the self-supervised learning phase in which the slow learner optimizes a Self-Supervised Learning (SSL) objective using unlabeled data from the episodic memory $\mathcal{M}$. Second, the supervised learning phase happens whenever a labeled sample arrives, which triggers the fast learner to first query the representation from the slow learner and adapt it to learn this sample. The incurred loss will be backpropagated into both learners for supervised knowledge consolidation. Additionally, the fast learner's adaptation is per-sample-based and does not require additional information such as the task identifiers. Note that DualNet uses the same episodic memory's budget as other methods to store the samples and their labels, but the slow learner only requires the samples while the fast learner uses both samples and their labels.

### 2.3 The Slow Learner

The slow learner is a standard backbone network $\phi$ trained to optimize an SSL loss, denoted by $\mathcal{L}_{SSL}$. As a result, any SSL objectives can be applied in this step. However, to minimize the additional computational resources while ensuring a general representation, we only consider the SSL loss that (i) does not require additional memory unit (such as the negative queue in MoCo [23]), (ii) does not always maintain an additional copy of the network (such as BYOL [21]), and (iii) does not use handcrafted pretext losses (such as RotNet [16] or JiGEN [6]). Therefore, we consider Barlow Twins [59], a recent state-of-the-art SSL method that achieved promising results with minimal computational overheads. Formally, Barlow Twins requires two views $\boldsymbol{M}^A$ and $\boldsymbol{M}^B$ by applying two different data transformations to a batch of images $\boldsymbol{M}$ sampled from the memory. The augmented data are then passed to the slow net $\phi$ to obtain two representations $\boldsymbol{Z}^A$ and $\boldsymbol{Z}^B$. The Barlow Twins loss is defined as:

$$\mathcal{L}_{\mathcal{BT}} \triangleq \sum_i (1 - \mathcal{C}_{ii})^2 + \lambda_{\mathcal{BT}} \sum_i \sum_{j \neq i} \mathcal{C}_{ij}^2, \tag{1}$$

where $\lambda_{\mathcal{BT}}$ is a trade-off factor, and $\mathcal{C}$ is the cross-correlation matrix between $\boldsymbol{Z}^A$ and $\boldsymbol{Z}^B$:

$$\mathcal{C}_{ij} \triangleq \frac{\sum_b z_{b,i}^A z_{b,j}^B}{\sqrt{\sum_B (z_{b,i}^A)^2} \sqrt{\sum_B (z_{b,j}^B)^2}} \tag{2}$$

with $b$ denotes the mini-batch index and $i, j$ are the vector dimension indices. Intuitively, by optimizing the cross-correlation matrix to be identity, Barlow Twins enforces the network to learn essential information that is invariant to the distortions (unit elements on the diagonal) while eliminating the redundancy information in the data (zero element elsewhere). In our implementation, we follow the

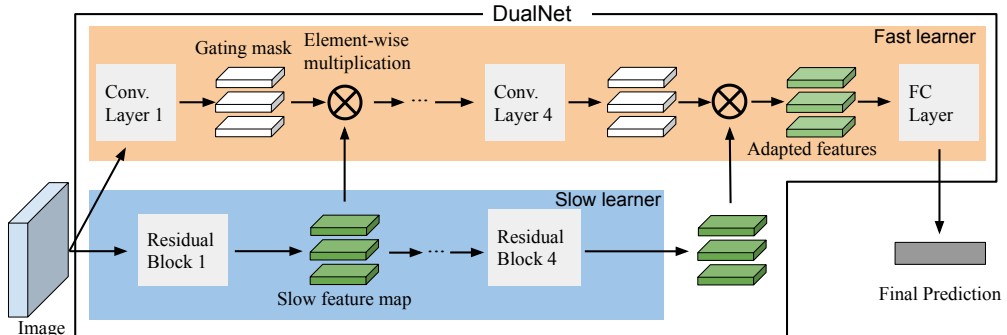

Figure 2: A demonstration of the slow and fast learners' interaction during the supervised learning or inference on a standard ResNet [22] backbone.

standard practice in SSL to employ a projector on top of the slow network's last layer to obtain the representations $\boldsymbol{Z}^A, \boldsymbol{Z}^B$. For supervised learning with the fast network, which will be described in Section 2.4, we use the slow network's last layer as the representation $\boldsymbol{Z}$.

In most SSL methods, the LARS optimizer [58] is employed for distributed training across many devices, which takes advantage of a large amount of unlabeled data. However, in continual learning, the episodic memory only stores a small number of samples, which are always changing because of the memory updating mechanism. As a result, the data distribution in the episodic memory always drifts throughout learning, and the SSL loss in DualNet presents different challenges compared to the traditional SSL optimization. Particularly, although the SSL objective in continual learning can be easily optimized using one device, we need to quickly capture the knowledge of the currently stored samples before the newer ones replace them. In this work, we propose to optimize the slow learner using the *Look-ahead* optimizer [61], which performs the following updates:

$$\tilde{\phi}_k \leftarrow \tilde{\phi}_{k-1} - \epsilon \nabla_{\tilde{\phi}_{k-1}} \mathcal{L}_{\mathcal{BT}}, \text{ with } \tilde{\phi}_0 \leftarrow \phi \text{ and } k = 1, \ldots, K \tag{3}$$

$$\phi \leftarrow \phi + \beta(\tilde{\phi}_K - \phi), \tag{4}$$

where $\beta$ is the Look-ahead's learning rate and $\epsilon$ is the Look-ahead's SGD learning rate. As a special case of $K = 1$, the optimization reduces to the traditional optimization of $\mathcal{L}_{\mathcal{BT}}$ using SGD. By performing $K > 1$ updates using a standard SGD optimizer, the look-ahead weight $\tilde{\phi}_K$ is used to perform a momentum update for the original slow learner $\phi$. As a result, the slow learner optimization can explore regions that are undiscovered by the traditional optimizer and enjoys faster training convergence [61]. Note that SSL focuses on minimizing the training loss rather than generalizing this loss to unseen samples, and the learned representation requires to be adapted to perform well on a downstream task. Therefore, such properties make the Look-ahead optimizer a more suitable choice over the standard SGD to train the slow learner.

Lastly, we emphasize that although we choose to use Barlow Twins as the SSL objective in this work, DualNet is compatible with any existing methods in the literature, which we will explore empirically in Section 4.3. Moreover, we can always train the slow learner in the background by optimizing Equation 1 synchronously with the continual learning of the fast learner, which we will detail in the following section.

## 2.4 The Fast Learner

Given a labeled sample $\{\boldsymbol{x}, y\}$, the fast learner's goal is utilizing the slow learner's representation to quickly learn this sample via an adaptation mechanism. In this work, we propose a novel context-free adaptation mechanism by extending and improving the channel-wise transformation [42, 43] to the general continual learning setting. Particularly, instead of generating the transformation coefficients based on the task-identifier, we propose to train the fast learner to learn such coefficients from the raw pixels in the image $\boldsymbol{x}$. Importantly, the transformation is *pixel-wise* instead of channel-wise to compensate for the missing input of task identifiers. Formally, let $\{h_i\}_{i=1}^L$ be the feature maps from the slow learner's layers on the image $\boldsymbol{x}$, e.g. $h_1, h_2, h_3, h_4$ are outputs from four residual blocks in ResNets [22], our goal is to obtain the adapted feature $h'_L$ conditioned on the image $\boldsymbol{x}$. Therefore, we

design the fast learner as a simple CNN with $L$ layers, and the adapted feature $h'_L$ is obtained as

$$m_l = g_{\boldsymbol{\theta},l}(h'_{l-1}), \text{ with } h'_0 = \boldsymbol{x} \text{ and } l = 1, \ldots, L \tag{5}$$

$$h'_l = h_l \otimes m_l, \quad \forall l = 1, \ldots, L, \tag{6}$$

where $\otimes$ denotes the element-wise multiplication, $g_{\boldsymbol{\theta},l}$ denotes the $l$-th layer's output from the fast network $\boldsymbol{\theta}$ and has the same dimension as the corresponding slow feature $h_l$. The final layer's transformed feature $h'_L$ will be fed into a classifier for prediction. Thanks to the simplicity of the transformation, the fast learner is light-weight but still can take advantage of the slow learner's rich representation. As a result, the fast network can quickly capture knowledge in the data stream, which is suitable for online continual learning. Figure 2 illustrates the fast and slow learners' interaction during the supervised learning or inference phase.

**The Fast Learner's Objective** To further facilitate the fast learner's knowledge acquisition during supervised learning, we also mix the current sample with previous data in the episodic memory, which is a form of experience replay (ER). Particularly, given the incoming labeled sample $\{\boldsymbol{x}, y\}$ and a mini-batch of memory data $\boldsymbol{M}$ belonging to a past task $k$, we consider the ER with a soft label loss [53] for the supervised learning phase as:

$$\mathcal{L}_{tr} = \text{CE}(\pi(\text{DualNet}(\boldsymbol{x}), y) + \frac{1}{|\boldsymbol{M}|} \sum_{i=1}^{|\boldsymbol{M}|} \text{CE}(\pi(\hat{y}_i), y_i) + \lambda_{tr} D_{\text{KL}} \left( \pi\left(\frac{\hat{y}_i}{\tau}\right) \middle\| \pi\left(\frac{\hat{y}_k}{\tau}\right) \right), \tag{7}$$

where CE is the cross-entropy loss , $D_{\text{KL}}$ is the KL-divergence, $\hat{y}$ is the DualNet's prediction, $\hat{y}_k$ is snapshot of the model's logit (the fast learner's prediction) of the corresponding sample at the end of task $k$, $\pi(\cdot)$ is the softmax function with temperature $\tau$, and $\lambda_{tr}$ is the trade-off factor between the soft and hard labels in the training loss. Similar to [43, 5], Equation 7 requires minimal additional memory to store the soft label $\hat{y}$ in conjunction with the image $\boldsymbol{x}$ and the hard label $y$.

## 3 Related Work

### 3.1 Continual learning

The CLS theory has inspired many existing continual learning methods in different settings [41, 12, 27], which can be broadly categorized into *two* groups. First, *dynamic architecture* methods aims at having a separate subnetwork for each task, thus eliminating catastrophic forgetting to a great extend. The task-specific network can be identified simply allocating new parameters [50, 57, 32], finding a configuration of existing blocks or activations in the backbone [17, 52], or generating the whole network conditioning on the task identifier [56]. While achieving strong performance, they are often expensive to train and do not work well on the online setting [8] because of the lack of knowledge transfer mechanism across tasks. In the second category of *fixed architecture* methods, learning is regularized by employing a memory to store information of previous tasks. In *regularization-based* methods, the memory stores the previous parameters and their importance estimations[29, 60, 1, 49], which regulates training of newer tasks to avoid changing crucial parameters of older tasks. Recent works have demonstrated that the *experience replay* (ER) principle [33] is an effective approach and its variants [36, 8, 48, 34, 53, 5] have achieved promising results. Notably, MER [48] extends the Reptile algorithm [38] for continual learning. Although achieving promising results on simple datasets such as MNIST, MER was later shown to be outperformed by the standard ER strategy on more challenging benchmarks based on CIFAR and miniImageNet [9]. Recently, CTN [43] was proposed to bridge the gap between the two approaches by having a fixed backbone network that can model task-specific features via the controller that models higher task-level information.

We argue that most existing methods fail to capture the CLS theory's fast and slow learning principle by coupling both representation types into one backbone network. Existing CLS-inspired method of FearNet [27] assumes having a powerful pre-trained representation and focuses on the memory management strategy. In contrast, DualNet explicitly maintains two separate systems, which facilitates slow representation learning to support generalization across tasks while allowing efficient and fast knowledge acquisition during continual learning.

### 3.2 Representation Learning for Continual Learning

Representation learning has been an important research field in Machine Learning and Deep Learning [16, 4]. Recent works demonstrated that a general representation could transfer well to finetune

on many downstream tasks [39], or generalize well under limited training samples [18]. For continual learning, extensive efforts have been devoted to learning a generic representation that can alleviate forgetting while facilitating knowledge transfer. The representation can be learned either by supervised learning [46], unsupervised learning [20, 40, 44], or meta (pre-)training [26, 24]. While unsupervised and meta training have shown promising results on simple datasets such as MNIST and Omniglot, they lack the scalability to real-world benchmarks. In contrast, our DualNet decouples the representation learning into the slow learner, which is scalable in practice by training synchronously with the supervised learning phase. Moreover, our work incorporates self-supervised representation learning into the continual learning process and does not require any pre-training steps.

## 3.3 Feature Adaptation

Feature adaptation allows the feature to quickly change and adapt [42]. Existing continual learning methods have explored the use of task identifiers [43, 56, 52] or the memory data [24] to support the fast adaptation. While the task identifier context is powerful since it provides additional information regarding the task of interest, such approaches are limited to the task-aware setting or require inferring the underlying task, which can be challenging in practice. On the other hand, data-based context conditioning is useful in incorporating information of similar samples to the current query and has found success beyond continual learning [18, 47, 15, 45]. However, we argue that naively adopting this approach is not practical for real-world continual learning because the model always performs the full forward/backward computation for each query instance, which reduces the inference speed and defeats the purpose of fast adaptation. Moreover, the predictions are not deterministic because of the dependency on the data chosen for finetuning. For DualNet, feature adaptation plays an important role in the interaction between the fast and slow learners. We address the limitations of existing techniques by developing a novel mechanism that allows the fast learner to efficiently utilize the slow representation without additional information about the task identifiers.

# 4 Experiments

Throughout this section, we compare DualNet against competitive continual learning approaches with a focus on the online scenario [36]. Our goal of the experiments is to investigate the following hypotheses: (i) DualNet's representation learning is helpful for continual learning; (ii) DualNet can continuously improve its performance via self-training in the background without any incoming data samples; and (iii) DualNet presents a general framework to unify representation learning and continual learning seamlessly and is robust to the choice of the self-supervised learning objective.

## 4.1 Experimental Setups

Our experiments follow the online continual learning under both the task-aware and task-free settings.

**Benchmarks** We consider the "Split" continual learning benchmarks constructed from the mini-ImageNet [54] and CORE50 dataset [35] with three validation tasks and 17, 10 continual learning tasks, respectively. Each task is created by randomly sampling without replacement five classes from the original dataset. For the task-aware (TA) protocol, the task identifier is available, and only the corresponding classifier is selected for evaluation. In contrast, the task-identifiers are not given in the task-free (TF) protocol, and the models have to predict all classes observed so far. We run the experiments five times and report the averaged accuracy of all tasks/classes at the end of training [36] (ACC), the forgetting measure [7] (FM), and the learning accuracy (LA) [48].

**Baselines** We compare our DualNet with a suite of state-of-the-art continual learning methods. First, we consider ER [9], a simple experience replay method that works consistently well across benchmarks. Then we include DER++ [5], an ER variant that augments ER with a $\ell_2$ loss on the soft labels. We also compare with CTN [43] a recent state-of-the-art method on the online task-aware setting. For all methods, the hyper-parameters are selected by performing grid-search on the cross-validation tasks.

**Architecture** We use a full ResNet18 [22] as the backbone in all experiments. In addition, we construct the DualNet's fast learner as follows: the fast learner has the same number of convolutional layers as the number of residual blocks in the slow learners. A residual block and its corresponding fast learner's layer will have the same output dimensions. With this configuration, the fast learner's architecture is uniquely determined by the slow learner's network. Lastly, all networks in our experiments are trained from scratch.

Table 1: Evaluation metrics on the Split miniImageNet and CORE50 benchmarks. All methods use an episodic memory of 50 samples per task in the TA setting, and 100 samples per class in the TF setting. The "Aug" suffix denotes using data augmentation during training

| Method | Split miniImageNet-TA | | | Split miniImageNet-TF | | |
|---|---|---|---|---|---|---|
| | ACC($\uparrow$) | FM($\downarrow$) | LA($\uparrow$) | ACC($\uparrow$) | FM($\downarrow$) | LA($\uparrow$) |
| ER | 58.24±0.78 | 9.22±0.78 | 65.36±0.71 | 25.12±0.99 | 28.56±1.10 | 49.04±1.56 |
| ER-Aug | 59.80±1.51 | 4.68±1.21 | 58.94±0.69 | 27.94±2.44 | 29.36±3.23 | 54.02±1.02 |
| DER++ | 62.32±0.78 | 7.00±0.81 | 67.30±0.57 | 27.16±1.99 | 34.56±2.48 | 59.54±1.53 |
| DER++-Aug | 63.48±0.98 | 4.01±1.21 | 62.17±0.52 | 28.26±1.81 | 36.70±1.85 | 62.70±0.41 |
| CTN | 65.82±0.59 | **3.02±1.13** | 67.43±1.37 | N/A | N/A | N/A |
| CTN-Aug | 68.04±1.23 | 3.94±0.98 | 69.84±0.78 | N/A | N/A | N/A |
| DualNet | **73.20±0.68** | 3.86±1.01 | **74.12±0.12** | **36.86±1.36** | **28.63±2.26** | **63.46±1.97** |

| Method | CORE50-TA | | | CORE50-TF | | |
|---|---|---|---|---|---|---|
| | ACC($\uparrow$) | FM($\downarrow$) | LA($\uparrow$) | ACC($\uparrow$) | FM($\downarrow$) | LA($\uparrow$) |
| ER | 41.72±1.30 | 9.10±0.80 | 48.18±0.81 | 21.80±0.70 | 14.42±1.10 | 33.94±1.49 |
| ER-Aug | 44.16±2.05 | 5.72±0.02 | 47.83±1.61 | 25.34±0.74 | 15.28±0.63 | 37.94±0.91 |
| DER | 46.62±0.46 | 4.66±0.46 | 48.32±0.69 | 22.84±0.84 | 13.10±0.40 | 34.50±0.81 |
| DER++-Aug | 45.12±0.68 | 5.02±0.98 | 47.67±0.08 | 28.10±0.80 | 10.43±2.10 | 36.16±0.19 |
| CTN | 54.17±0.85 | 5.50±1.10 | 55.32±0.34 | N/A | N/A | N/A |
| CTN-Aug | 53.40±1.37 | 6.18±1.61 | 55.40±1.47 | N/A | N/A | N/A |
| DualNet | **57.64±1.36** | **4.43±0.82** | **58.86±0.66** | **38.76±1.52** | **8.06±0.43** | **40.00±1.67** |

**Training** In the supervised learning phase, all methods are optimized by the (SGD) optimizer **over one epoch** with mini-batch size 10 and 32 on the Split miniImageNet and CORE50 benchmarks respectively [43]. In the representation learning phase, we use the Look-ahead optimizer [61] to train the DualNet's slow learner as described in Section 2.3. We employ an episodic memory with *50 samples per task* and the Ring-buffer management strategy [36] in the task-aware setting. In the task-free setting, the memory is implemented as a reservoir buffer [55] with 100 samples per class. We simulate the synchronous training property in DualNet by training the slow learner with $n$ iterations using the episodic memory data before observing a mini-batch of labeled data.

**Data pre-processing** DualNet's slow learner follows the data transformations used in BarlowTwins [59]. For the supervised learning phase, we consider two settings. First, the standard data pre-processing of no data augmentation during both training and evaluation. Second, we also train the baselines with data augmentation for a fair comparison. However, we observe the data transformation in [59] is too aggressive; therefore, only implement the random cropping and flipping for the supervised training phase. In all settings, no data augmentation is applied during inference.

### 4.2 Results on Online Continual Learning Benchmarks

Table 1 reports the evaluation metrics on the CORE50 and Split miniImageNet benchmarks, where we omit CTN's performance on the task-free setting since it is strictly a task-aware method. Our DualNet's slow learner optimizes the Barlow Twins objective for $n = 3$ iterations between every incoming mini-batch of labeled data. Data augmentation creates more samples to train the models and provides improvements to all baselines on all benchmarks. Consistent with previous studies, we observe that DER++ performs slightly better than ER thanks to its soft-label loss. Similarly, CTN can perform better than both ER and DER++ because of its ability to model task-specific features. Overall, our DualNet consistently outperforms other baselines by a large margin, even with the data augmentation propagated to their training. Specifically, DualNet is more resistant to catastrophic forgetting (lower FM) while greatly facilitating knowledge transfer (higher LA), which results in better overall performance indicated by higher ACC. Since our DualNet has a similar supervised objective as DER++, this result shows that the DualNet's decoupled representations and its fast adaptation mechanism are beneficial to continual learning.

### 4.3 Ablation Study of Slow Learner Objectives and Optimizers

We now study the effects of the slow learner's objective and optimizer on the final performance of DualNet. We consider several objectives to train the slow learner. First, we consider the *classification*

Table 2: DualNet's performance under different slow learner objective and optimizers on the Split miniImageNet-TA benchmark

| Objective | SGD | | | Look-ahead | | |
|---|---|---|---|---|---|---|
| | ACC($\uparrow$) | FM($\downarrow$) | LA($\uparrow$) | ACC($\uparrow$) | FM($\downarrow$) | LA($\uparrow$) |
| Barlow Twins | 64.20$\pm$2.37 | 4.79$\pm$1.19 | 64.83$\pm$1.67 | **73.20$\pm$0.68** | **3.86$\pm$1.01** | **74.12$\pm$0.12** |
| SimCLR | **71.49$\pm$1.01** | **4.23$\pm$0.46** | **72.64$\pm$1.20** | 72.13$\pm$0.44 | 4.13$\pm$0.52 | 73.09$\pm$0.16 |
| SimSiam | 70.55$\pm$0.98 | 4.93$\pm$1.31 | 71.90$\pm$0.65 | 71.94$\pm$0.64 | 4.21$\pm$0.28 | 72.93$\pm$0.38 |
| BYOL | 69.76$\pm$2.12 | **4.23$\pm$1.41** | 70.33$\pm$0.87 | 71.73$\pm$0.47 | **3.96$\pm$0.62** | 72.06$\pm$0.28 |
| Classification | 68.50$\pm$1.67 | 5.53$\pm$1.67 | **72.93$\pm$1.10** | 70.96$\pm$1.08 | 6.33$\pm$0.28 | 73.92$\pm$1.14 |

Table 3: Performance of DualNet with different self-supervised learning iterations $n$ on the Split miniIMageNet benchmarks

| $n$ | Split miniImageNet-TA | | | Split miniImageNet-TF | | |
|---|---|---|---|---|---|---|
| | ACC($\uparrow$) | FM($\downarrow$) | LA($\uparrow$) | ACC($\uparrow$) | FM($\downarrow$) | LA($\uparrow$) |
| 1 | 72.26$\pm$0.71 | 3.80$\pm$0.69 | 73.16$\pm$1.51 | 33.40$\pm$3.28 | 32.86$\pm$3.06 | 63.96$\pm$0.53 |
| 3 | 73.20$\pm$0.68 | 3.86$\pm$1.01 | 74.12$\pm$0.12 | 36.86$\pm$1.36 | 28.63$\pm$2.26 | 63.46$\pm$1.97 |
| 10 | 74.10$\pm$1.03 | 3.67$\pm$0.80 | 74.68$\pm$0.52 | 36.43$\pm$1.73 | 30.92$\pm$2.16 | **65.33$\pm$0.52** |
| 20 | **74.53$\pm$1.18** | **3.48$\pm$0.45** | **75.60$\pm$0.65** | **38.56$\pm$1.91** | **27.96$\pm$1.71** | 64.06$\pm$0.67 |

*loss* to train the slow net, which reduces DualNet's representation learning to supervised learning. Second, we consider various contrastive SSL losses, including SimCLR [10], SimSiam [11], and BYOL [21]. We consider the Split miniImageNet-TA and TF benchmark with 50 memory slots per task and optimize each objective using the SGD and Look-ahead optimizers. Table 2 reports the result of this experiment. In general, we observe that SSL objectives achieve a better performance than the classification loss. Moreover, the Look-ahead optimizer consistently improves the performances on all objectives compared to the SGD optimizer. This result shows that our DualNet's design is general and can work well on different slow learner's objectives. Interestingly, we also observe that when using the Look-ahead optimizer, the Barlow Twins loss achieves better performance than the remaining objective, which is also the case for the supervised training [59]. Therefore, we expect DualNet to improve its performance with a more powerful and suitable slow learning objective.

### 4.4 Ablation Study of Self-Supervised Learning Iterations

We now investigate DualNet's performances with different SSL optimization iterations $n$. Small values of $n$ indicate there is little to no delay of labeled data from the continuum, and the fast learner has to query the slow learner's representation continuously. On the other hand, larger $n$ simulate the situations where labeled data are delayed, which allows the slow learner to train its SSL objective for more iterations between each query from the fast learner. In this experiment, we gradually increase the SSL training iterations between each supervised update by varying from $n = 1$ to $n = 20$. We run the experiments on both the Split miniImageNet benchmarks under the TA and TF settings. Table 3 reports the result of this experiment. Interestingly, even with only one SSL training iteration ($n = 1$), DualNet still obtains competitive performance and outperforms existing baselines. As more iterations are allowed, DualNet consistently reduces forgetting and facilitates knowledge transfer, which results in a better overall performance.

### 4.5 Ablation Study of DualNet's Fast Learner

DualNet's introduces an additional fast learner on top of the standard backbone used as the slow learner. In this experiment, we investigate the contribution of the fast learner to DualNet's overall performance on the Split miniImageNet on both the TA and TF settings. We compare the full DualNet against a variant that only employs a slow learner and report the results in Table 4. We can see that the slow learner variant binds both types of representation into the same backbone and performs significantly worse than the original DualNet on both scenarios. This result corroborates with our motivation in Section 1 that it is more beneficial to separate the two self-supervised and supervised representations into two distinct systems.

Table 4: Evaluation of DualNet's slow learner on the Split miniImageNet TA and TF benchmarks

| DualNet | Split miniImageNet-TA | | | Split miniImageNet-TF | | |
|---|---|---|---|---|---|---|
| | ACC($\uparrow$) | FM($\downarrow$) | LA($\uparrow$) | ACC($\uparrow$) | FM($\downarrow$) | LA($\uparrow$) |
| Slow Learner | **73.20$\pm$0.68** | **3.86$\pm$1.01** | **74.12$\pm$0.12** | **36.86$\pm$1.36** | **28.63$\pm$2.26** | **63.46$\pm$1.97** |
| Slow+Fast Leaners | 68.33$\pm$0.57 | 5.12$\pm$0.78 | 69.20$\pm$0.32 | 27.30$\pm$0.25 | 34.60$\pm$1.12 | 59.70$\pm$1.26 |

Table 5: Evaluation metrics on the Split miniImageNet-TA benchmarks under the semi-supervised setting, where $\rho$ denotes the fraction of data are labeled

| Method | $\rho = 10\%$ | | | $\rho = 25\%$ | | |
|---|---|---|---|---|---|---|
| | ACC($\uparrow$) | FM($\downarrow$) | LA($\uparrow$) | ACC($\uparrow$) | FM($\downarrow$) | LA($\uparrow$) |
| ER | 41.66$\pm$2.72 | 6.80$\pm$2.07 | 42.33$\pm$1.51 | 50.13$\pm$2.19 | 6.76$\pm$1.51 | 51.90$\pm$2.16 |
| DER++ | 44.56$\pm$1.41 | 4.55$\pm$0.66 | 43.03$\pm$0.71 | 51.63$\pm$1.11 | 6.03$\pm$1.46 | 52.36$\pm$0.55 |
| CTN | 49.80$\pm$2.66 | 3.96$\pm$1.16 | 47.76$\pm$0.99 | 55.90$\pm$0.86 | 3.84$\pm$0.32 | 55.69$\pm$0.98 |
| DualNet | **54.03$\pm$2.88** | **3.46$\pm$1.17** | **49.96$\pm$0.17** | **62.80$\pm$2.40** | **3.13$\pm$0.99** | **59.60$\pm$1.87** |

## 4.6 Results of the Semi-Supervised Continual Learning Setting

In real-world continual learning scenarios, there exist abundant unlabeled data, which are costly and even unnecessary to label entirely. Therefore, a practical continual learning system should be able to improve its representation using unlabeled samples while waiting for the labeled data. To test the performance of existing methods on such scenarios, we create a *semi-supervised continual learning* benchmark, where the data stream contains both labeled and unlabeled data. For this, we consider the Split miniImageNet-TA benchmark but provide labels randomly to a fraction ($\rho$) of the total samples, which we set to be $\rho = 10\%$ and $\rho = 25\%$. The remaining samples are unlabeled and cannot be processed by the baselines we considered so far. In contrast, such samples can go directly to the DualNet's slow learner to improve its representation while the fast learner stays inactive. Other configurations remain the same as the experiment in Section 4.2.

Table 5 shows the results of this experiment. Under the limited labeled data regimes, the results of ER and DER++ drop significantly. Meanwhile, CTN can still maintain competitive performances thanks to additional information from the task identifiers, which remains untouched. On the other hand, DualNet can efficiently leverage the unlabeled data to improve its performance and outperform other baselines, even CTN. This result demonstrates DualNet's potential to work in a real-world environment, where labeled data can be delayed or even unlabeled.

Due to space constraints, we refer to the supplementary materials for DualNet's pseudo-code, additional results, experiments settings such as dataset summary, evaluation metrics, hyper-parameter configurations, and further discussions.

## 5 Conclusion

In this paper, we proposed DualNet, a novel paradigm for continual learning inspired by the fast and slow learning principle of the Complementary Learning System theory from neuroscience. DualNet comprises two key learning components: (i) a slow learner that focuses on learning a general and task-agnostic representation using the memory data; and (ii) a fast learner focuses on capturing new supervised learning knowledge via a novel adaptation mechanism. Moreover, the fast and slow learners complement each other while working synchronously, resulting in a holistic continual learning method. Our experiments on two challenging benchmarks demonstrate the efficacy of DualNet. Lastly, extensive and carefully designed ablation studies show that DualNet is robust to the slow learner's objectives, scalable with more resources, and applicable to the semi-supervised continual learning setting.

Our DualNet presents a general continual learning framework that can enjoy great scalability to real-world continual learning scenarios. However, additional computational cost incurred to train the slow learner continuously should be properly managed. Moreover, applications to specific domains should take into account the inherent challenges. Lastly, we adopt the contrastive SSL approach to train the DualNet's slow learner in this work. Future work includes designing a slow objective tailored for continual learning.

## Acknowledgement

The first author is supported by the SMU PGR scholarship. We thank the anonymous Reviewers for helpful discussions during the submission of this work.

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
