# DualNet: Continual Learning, Fast and Slow
## Supplementary Materials

**Quang Pham**[1], **Chenghao Liu**[2], **Steven C.H. Hoi** [1,2]

[1] Singapore Management University
`hqpham.2017@smu.edu.sg`
[2] Salesforce Research Asia
`{chenghao.liu, shoi}@salesforce.com`

The structure of this document is organized as follows. In Section A, we provide the pseudo-code of DualNet. Section B provides additional details of our experiments, including datasets, evaluation metrics, baselines' descriptions, additional results, and hyper-parameter settings. Lastly, we discuss the potential impacts of DualNet in Section C.

35th Conference on Neural Information Processing Systems (NeurIPS 2021).

# A Algorithm

We provide the pseudo-code to train DualNets in Algorithm 1.

---
**Algorithm 1:** Psuedo-code to train DualNets.
---

1 **Algorithm** `TrainDualNet`$(\boldsymbol{\theta}, \boldsymbol{\phi}, \mathcal{D}_{1:T}^{tr})$
    **Require:** slow learner $\boldsymbol{\phi}$, fast learner $\boldsymbol{\theta}$, episodic memory $\mathcal{M}$, inner updates $N$, Look-ahead
          inner updates $K$ (for Look-ahead)
    **Init:** $\boldsymbol{\theta}, \boldsymbol{\phi}, \mathcal{M} \leftarrow \varnothing$
2     **for** $t \leftarrow 1$ **to** $T$ **do**
3         **for** $j \leftarrow 1$ **to** $n_{batches}$ **do**                 // Receive the dataset $D_t^{tr}$ sequentially
4             Receive a mini batch of data $\mathcal{B}_j$ from $\mathcal{D}_t^{tr}$
5             $\boldsymbol{x}^*, y^* \leftarrow$ Random sampling from $\mathcal{B}_j$   // Sampling for the semantic memory
6             $\mathcal{M} \leftarrow$ **MemoryUpdate**$(\mathcal{M}, \mathcal{B}_j)$            // Update the episodic memory
7             **for** $j \leftarrow 1$ **to** $\infty$ **do**           // Train the slow learner synchronously
8                Train the slow learner using the Look-ahead procedure
9             **for** $n \leftarrow 1$ **to** $N$ **do**           // Train the fast learner synchronously
10                $\boldsymbol{M}_n \leftarrow$ Sample$(\mathcal{M})$
11                $\mathcal{B}_n \leftarrow \boldsymbol{M}_n \cup \mathcal{B}_j$
12                SGD update the slow learner: $\boldsymbol{\phi} \leftarrow \boldsymbol{\phi} - \nabla_{\boldsymbol{\phi}} \mathcal{L}_{tr}(\mathcal{B}_n)$
13                SGD update the fast learner $\boldsymbol{\theta} \leftarrow \boldsymbol{\theta} - \nabla_{\boldsymbol{\theta}} \mathcal{L}_{tr}(\mathcal{B}_n)$
14         $\mathcal{M}_t^{em} \leftarrow \mathcal{M}_t^{em} \cup \{\pi(\hat{y}/\tau)\}$
15     **return** $\boldsymbol{\theta}, \boldsymbol{\phi}$

1 **Procedure** `Look-ahead`$(\boldsymbol{\phi}, \mathcal{M})$
2     $\tilde{\phi}_0 \leftarrow \boldsymbol{\phi}$
3     **for** $k \leftarrow 1$ **to** $K - 1$ **do**
4         $\boldsymbol{M}_k \leftarrow$ Sample$(\mathcal{M})$
5         Obtains two views of $\boldsymbol{M}_k : \boldsymbol{M}_k^A, \boldsymbol{M}_k^B$
6         Calculate the Barlow Twins loss: $\mathcal{L}_{\mathcal{BT}}(\tilde{\phi}_k, \boldsymbol{M}_k^A, \boldsymbol{M}_k^B)$
7         SGD update the slow learner: $\tilde{\phi_{k+1}} \leftarrow \tilde{\phi}_k - \epsilon \nabla_{\phi_k} \mathcal{L}_{\mathcal{BT}}$
8     Look-ahead update the slow learner: $\boldsymbol{\phi} \leftarrow \boldsymbol{\phi} + \beta(\tilde{\phi}_K - \boldsymbol{\phi})$
9     **return** $phi$

---

# B Experiment Details

## B.1 Datasets and Evaluation Metrics

We summary the datasets used in our experiments in Table 1.

Table 1: Summary of datasets used in our experiments, including number of classes, number of data, dimension with and without data augmentation

| Dataset | Classes | Train | Test | Dim. | Dim. with Aug. |
|---|---|---|---|---|---|
| miniImageNet [11] | 100 | 50,000 | 10,000 | $3 \times 84 \times 84$ | $3 \times 128 \times 128$ |
| CORe50 [5] | 50 | 119,894 | 44,971 | $3 \times 84 \times 84$ | $3 \times 128 \times 128$ |

We consider three standard metrics to evaluate the methods [8]: ACC [6] (higher is better), FM [2] (lower is better), and LA [10] (higher is better). We define $a_{i,j}$ as the model's accuracy evaluated on the testing data of task $j$ after it is trained on the last mini-batch of task $i$. The above metrics are defined as:

- **Average Accuracy (ACC):**

$$\text{ACC} = \frac{1}{T} \sum_{i=1}^{T} a_{T,i}. \tag{1}$$

- **Forgetting Measure (FM):**

$$\text{FM} = \frac{1}{T-1} \sum_{j=1}^{T-1} \max_{l \in \{1,\ldots T-1\}} a_{l,j} - a_{T,j}. \tag{2}$$

- **Learning Accuracy (LA):**

$$\text{LA} = \frac{1}{T} \sum_{i=1}^{T} a_{i,i}. \tag{3}$$

We run each experiment five times with the same task-order but different initialization and report the mean, standard deviation of each model.

## B.2   Baselines

We provide a brief description of the baselines considered in this work as follows:

- **ER** [3]: a standard experience replay that interleaves the current mini-batch with a mini-batch randomly sampled from the memory in each update.

- **DER++** [1]: a variant of ER with an additional regularizer using the $\ell_2$ distance between the current and past model's logits.

- **CTN** [8]: a static architecture method that can model task-specific features via a light-weight controller module.

- **Offline**: an upper bound model that performs multitask training on all data. Note that this model does not follow the continual learning setting. We implement the offline model by training the network three epochs over all data of all tasks.

For each method, we also include a complementary variant trained using data augmentation for a fair comparison with our DualNet. We use the same transformations from [12] includes: random cropping, resizing to $84 \times 84$, horizontal flip, color jittering, converting to greyscale, Gaussian blurring. The augmentation parameters are kept the same as [12, 4]. However, in the task-free setting, we observed that such this data augmentation was too aggressive in the supervised learning phase, which made training unstable, and often diverged across many runs. Therefore, we only use the random cropping and resize transformations for the baselines. For simplicity, we do not apply data augmentation to DualNet's supervised training.

## B.3   Additional Experiment Results in The Batch Setting

We develop DualNet with a focus on the online continual learning setting [6]. In this experiment, we explore DualNet's behaviors in the batch continual learning setting [9] where a learner is allowed to revisit the current task's data for many epochs before moving to the next task. We compare DualNet with DER++ [1] trained in the batch setting of 20 epochs per task with early stopping. Table 2 reports the results of this experiment. We observe that batch training allows the both methods to learn individual tasks well, results in a better LA. At the same time, batch training is also prone to catastrophic forgetting and results in higher FM values. However, the improvement in LA is greater than FM, which results in better overall performance (ACC). We also observe that the gap between DualNet and DER++ is not as significant as the online settings. The reason is that DualNet's slow representation facilitates fast knowledge acquisition on data streams, where the model is not allowed to revisit past samples. However, this condition is much relaxed in the batch setting because the model is allowed to train for more epochs to solve the current task.

Table 2: Comparison between DualNet and DER++ trained in the batch setting

| Method | Split miniImageNet-TA | | | Split miniImageNet-TF | | |
|---|---|---|---|---|---|---|
| | ACC | FM | LA | ACC | FM | LA |
| DER++ | 71.96±1.36 | 7.26±1.50 | 78.23±0.85 | 27.50±1.83 | 37.52±2.83 | 62.30±0.43 |
| DualNet | **75.20±0.09** | **6.08±0.23** | **78.99±0.10** | **38.26±1.10** | **31.83±0.93** | **66.63±0.61** |

Table 3: Performance of the Offline model under different configuration on the Split miniImageNet-TA benchmark, * denotes the method is trained in the continual learning setting - for reference

| Architecture | Method | Loss | Data Aug | ACC |
|---|---|---|---|---|
| Fast+Slow nets | DualNet* | SL+SSL | Yes | 73.20±0.68 |
| | Offline | SL | No | 75.83±1.07 |
| | Offline | SL | Yes | **77.63±0.48** |
| | Offline | SL+SSL | Yes | **77.98±0.16** |
| Slow net | Offline | SL | No | 71.15±2.95 |
| | Offline | SL | Yes | 75.46±0.97 |

## B.4 DualNet's Upper Bound

In our work, there are three factors affecting DualNet's upper bound model: (i) model architecture: slow net (standard backbone) versus fast and slow nets (DualNet); (ii) training loss: supervised learning loss (SL) or supervised and self-suppervised learning losses (SL+SSL); and (iii) data augmentation. As a result, we believe that an upper bound of DualNet is a model having all three factors as DualNet (has fast and slow learners, optimized both SL and SSL losses with data augmentation) and is trained offline. The offline model has access to all tasks' data to simultaneously optimizes both the SL and SSL losses, which are backpropagated through both learners. Here we consider the offline model trained up to five epochs.

We explore different combinations of the aforementioned factors to train an Offline model on the Split miniImageNet-TA benchmark and report the result in Table 3. Note that the configuration of Slow Net + Offline + SL + no data augmentation is the previous result reported in [8]. Our argued upper bound for DualNet has the following configuration: Fast + Slow nets + Offline + SL + SSL + data augmentation. The result confirms the upper bound of DualNet. Moreover, in the offline training with all data, the SSL only contributes a minor improvement to the SL. However, in continual learning, SSL is more beneficial because its representation does not depend on the class label, and therefore more resistant to catastrophic forgetting when old task data is limited.

## B.5 Hyper-parameter Setting

We implement all experiments in this work using the Pytorch [7] framework on a single K80 GPU. We provide the hyper-parameters values of each method we considered. For consistency, we use the same notations with respect to the original papers.

- ER
  - Learning rate: 0.03 (all benchmarks)
  - Replay batch size: 10 (all benchmarks)
  - Number of gradient updates: 2 (all TA experiments), 3 (all TF experiments)
- DER++
  - Learning rate: 0.03 (all benchmarks)
  - Replay batch size: 10 (all benchmarks)
  - Trade-off strength between soft and hard labels: 0.1 (all experiments)
  - Number of gradient updates: 2 (all TA experiments), 3 (all TF experiments)
- CTN

- Inner learning rate $\alpha$: $0.01$ (all benchmarks)
- Outer learning rate $\beta$: $0.05$ (all benchmarks)
- Regularization strength $\lambda$: $100$ (all benchmarks)
- Temperature $\tau$: $5$ (all benchmarks)
- Replay batch size: $64$ (all benchmarks)
- Number of inner and outer updates: $2$ (all benchmarks)
- Semantic memory size in percentage of total memory: $20\%$ (all benchmarks)
- DualNet
    - Slow learner's SGD learning rate: $3e-4$ (Split miniImageNet bencharks), $1e-4$ (CORE50 benchmarks)
    - Slow learner's Look-ahead learning rate: $0.5$ (all benchmarks)
    - Fast learner's learning rate: $0.03$ (all benchmarks)
    - Barlow Twins's trade-off term $\lambda_{\mathcal{BT}} : 2e-3$ (all benchmarks)
    - Fast learner's trade-off term $\lambda_{train} : 2.0$ (all benchmarks)
    - Soft label loss temperature $\tau : 2.0$ (all TA benchmarks), $10.0$ (all TF benchmarks)
    - Replay batch size: $10$ (Split miniImageNet bencharks), $32$ (CORE50 benchmarks)

We perform grid search for hyper-parameter cross-validation on the **three validation tasks**. The grid for each hyper-parameter is:

- Learning rate, including inner, outer (CTN) and DualNet fast and learners learning rates: $[0.0001, 0.0003, 0.001, 0.003, 0.01, 0.03, 0.05, 0.1, 0.3, 0.5]$
- Replay batch size: $[10, 32, 64, 128]$
- Temperature $\tau$: $[1, 2, 5, 10]$
- Regularization strength
    - $\lambda$ (CTN): $[0.1, 0.5, 1, 2, 10, 100]$
- Semantic memory size in percentage of total memory (CTN): $[10\%, 20\%, 30\%, 40\%]$

## C   Discussion

This section discusses DualNet's broader impacts, its limitations and potentials.

**Broader Impact**   Our research advances *continual learning* that learn continuously from a stream of tasks and data, which is a hallmark challenge in Artificial Intelligence. Enabling continual learning with neural networks has a large impact on a wide range of applications. It may advance our understanding of the human brain in neuroscience because human is a continual natural learner. Despite extensive efforts and recent success, existing continual methods are still far from practical usage and lack the ability to tackle the challenges during deployment. Our research proposes a novel continual learning framework suitable for deployment. Particularly, real-world deployment can often encounter scenarios where labeled data are delayed or mixed with unlabeled data. Our work have demonstrated that DualNet is readily applicable to such settings while existing works cannot.

**Limitations and Potentials**   Because the DualNet's slow learner can always be trained in the background, it incurs computational costs that need to be properly managed. For large-scale systems, such additional computations can increase infrastructural costs substantially. Therefore, it is important to manage the slow learner to balance between performance and computational overheads. In addition, implementing DualNet to specific applications requires additional considerations to address its inherent challenges. For example, medical image analysis applications may require paying attention to specific regions in the image or considering the data imbalance. However, since we demonstrate the efficacy of DualNet in general settings, such properties are not considered. In practice, it would be more beneficial to capture such domain-specific information to achieve better results. For general applications of DualNet, we also expect that a more suitable objective to train the slow learner can further improve the results.

**Conclusion**  Overall, inspired by neuroscience, our research proposes DualNet, a novel continual learning method suitable for practical deployment. We wish our work can encourage more research on continual learning to better understand machine intelligence and human intelligence.