# OpenReview forum: "DualNet: Continual Learning, Fast and Slow"
_NeurIPS.cc/2021/Conference — NeurIPS 2021 Poster_

### Official Review · Reviewer_qVwg · 2021-06-30

**Rating:** 7
**Confidence:** 4

**Summary:**

This work proposes an Architectural-based method for online continual learning, suitable both for the Task-aware and Task-free scenarios. Inspired by biologic suggestions, the author separate a continual learner in a slow network (which learns generic representations in a contrastive fashion) and a fast network (which utilises the feature generated by the former to produce a classification response).
Experiments show state-of-the-art accuracy in the presented setting.

**Limitations And Societal Impact:**

This is theoretical Continual Learning work that only refers to in-the-lab classification problems. I believe that the limitations of such a work are implicitly defined by the specific evaluation setting detailed in sec. 4.1. Discussion of societal impacts are not applicable due to the abstract nature of this work.

**Main Review:**

Strong points of this work:
- I find it clearly written and easy to understand (with all the information and the pseudo-code provided in the supplementary material, I feel confident I could re-implement this work from scratch);
- I find the idea to combine a separated representation learner with a lean feature adaptation model to be both novel (other works propose dual structures, but not in this way) and convincing.
- The results include a very up-to-date roster of competitors, although not necessarily fair as I explain below.

Weaknesses and suggestions:
- The main issue that could be brought up about this paper is that competitors are not as sample-efficient as the proposed method. If I understand correctly, in its slow-learning branch, DualNet can optimise memory data repeatedly between batches from the continual input stream, while ER and DER++ only apply one update for each input batch (as they are not necessarily designed with the online setting in mind - Buzzega et al. perform multiple epochs on the same task). It could be interesting to add more sample-efficient competitors, which should be especially effective in the online scenario, e.g. MER [Riemer et al, 2019 in the paper] and GSS [1]
- While I found the ablation section interesting, I feel that reproducing the same experiments in a task-free setting would make the results of Tables 2,3,4 much more interesting. In general, the task-aware setting is considered to be simplistic and less indicative of good performance w.r.t. the task-free one [1, 2].
- A very minor point, one of the first CL models that came to my mind while reviewing this work is Fearnet [3], which is not cited and also borrows from similar biological premises. However, the latter is not necessarily an up-to-date reference, so I only indicate it here as a suggestion.

The following points did not affect my judgment, but I feel that they should be fixed:
- The first citation of Kirkpatrick et al., 2017 at line 34 seems out of place, as EWC is not a method that uses an episodic memory.
- The citation style for this paper is not the one that is found in typical NeurIPS 2021 submissions, namely using increasing numbers between brackets [X].
- The following typos could be fixed:
line 158: e.g. seems to be followed by a slightly larger space (the space after the last period might need escaping)
Equation 7: what does index k denote?
line 191: missing space between estimations and the citations
line 255: there's a misplaced u character after the citation
Table 2: The results for Barlow Twins, (SGD, ACC) have /- rather than \plusminus
Table 3 caption: awarae -> aware
Table 4: \rho=10 should have a percentage
line 327: to label all of them --> to entirely label

[1] Aljundi et al., Gradient based sample selection for online continual learning, NeurIPS 2019
[2] Farquhar et al., Towards Robust Evaluations of Continual Learning, ICML workshop 2018
[3] Kemker et al., FearNet: Brain-Inspired Model for Incremental Learning, ICLR 2018


--- AFTER REBUTTAL ---

I thank the authors for engaging in the rebuttal process. I especially found the experiments provided as a response to reviewer RowY to be interesting and meaningful, so I strongly encourage the authors to include them in their final revision for this paper.
I confirm my previous rating and recommend acceptance of this work.

**Time Spent Reviewing:**

4

---

> ### Author Response · Authors · 2021-08-10
> **Responses to Reviewer qVwg**
>
> **Concern #1: sample efficiency in the baselines.**
>
> Please refer to our response to concerns #2 Reviewer RowY for the discussion regarding the sample efficiency.
> We couldn’t compare with GSS and MER during the rebuttal period because they take too long to train, which is also observed in [2]. Moreover, we also want to note that both GSS and MER have shown limited performances on more challenging datasets based on miniImageNet [2,3]. Therefore, we will try to include them in the final version.
>
> **Concern #2: repeat the ablation studies in the task-free setting.**
>
> Thanks for the suggestion. We agree with the Reviewer that replicating the current ablation studies in the task-free setting would be interesting, which we will include in the final version.
> In this rebuttal, please find our response to Concern #1, Reviewer RowY, where we conduct an ablation study of removing the fast learner and observe similar results on both the task-aware and task-free settings.
>
> **Concern #3: FearNet reference.**
>
> Thanks for the suggestion. A key difference between FearNet and our work is that FearNet focuses on replay and memory management strategies while assuming having a strong, fixed representation (FearNet uses a pre-trained model to extract the features). On the other hand, our work focuses on learning the representation during continual learning using a network initialized from scratch. We will include this discussion in the final version.
>
> **EWC is not a method that uses an episodic memory**
>
> Thank you for spotting this mistake, we will correct this in the final version.
>
> **Index k in Equation 7.**
>
> In index k refers to the k-th task and \hat{y}_k refers to the snapshot of the model’s logit of the corresponding sample at the end of the k-th task.
>
> **Typos and formatting.**
>
> Thank you again for carefully checking our paper. We will edit such mistakes in the final version of this work.
>
>
> [1] Aljundi, Rahaf, et al. "Online Continual Learning with Maximal Interfered Retrieval." Advances in Neural Information Processing Systems 32 (2019): 11849-11860.
>
> [2] Pietro Buzzega, Matteo Boschini, Angelo Porrello, Davide Abati, and Simone Calderara. Dark experience for general continual learning: a strong, simple baseline. In 34th Conference on Neural Information Processing Systems (NeurIPS 2020), 2020.
>
> [3] Chaudhry, Arslan, et al. "On tiny episodic memories in continual learning." arXiv preprint arXiv:1902.10486 (2019).

---

### Official Review · Reviewer_RowY · 2021-07-10

**Rating:** 7
**Confidence:** 4

**Summary:**

This paper introduces a model for continual learning based on example replay. Motivated by the Complementary Learning Systems theory in neuroscience, the authors propose to use, instead of a single backbone network, two different models: the slow and the fast learner. The slow learner is composed of a standard backbone architecture, such as ResNet, and it is optimized via self-supervised learning with examples from a memory buffer. The fast learner performs an adaptation of representations provided by the slow learner (based on masking) and embeds a classifier. The fast learner is supervised with class labels. Experiments and ablation studies are carried out on two datasets, namely miniImagenet and CORE-50, both in the presence and in the absence of task labels at inference time.

**Limitations And Societal Impact:**

Yes, there is a discussion in the supplementary material.

**Main Review:**

The paper seems solid and it is very well written.
Most equations seem correct and visuals help understanding the modeling choices.
The idea of exploiting self-supervised learning in continual learning is fairly new to my knowledge, and so it is the idea of using multiple networks with different objectives.
The experimental results seem really encouraging.

However, I have some doubts about the experimental setting and the validation of the proposed components that I think are worth raising:
1) Not all the components hereby introduced are properly validated. Specifically, the feature adaptation transforming representations from the slow learner to the ones of the fast learner (eq. 5 and 6) is also not properly validated. How can we be sure this step is necessary? I think a good baseline could be using features from the slow learner directly, without any adaptation, and applying a classifier on top of them.
2) I am unsure why the authors decide to optimize only one epoch per task. In my experience, that sort of constraint (unfortunately fairly popular in CL literature) introduces some sample efficiency problems and confounds them with catastrophic forgetting. As such, methods that are more sample efficient can perform better, but not because of their continual learning capabilities. Similar considerations are also discussed in the DER paper [1] (appendix, section F.3). In the context of this paper, this issue is even more serious as the training involves some optimization choices that might improve sample efficiency. Specifically: i) the look-ahead classifier, requiring more forward-backward updates for every example (boosting the performances significantly, Tab. 2) and ii) the fact that the self-supervised learning objective is optimized n times (with n=3 in the comparisons with state of the art).
This is further testified by the fact that DualNet outperforms Offline training, that is considered the upper bound in continual learning settings (Tab. 2 in supplementary material).
In summary: to enable a fair comparison, is the number of updates performed by DualNet the same as the one allowed to competitors such as ER, DER, or CTN? Or even better, are the positive results consistent in longer optimizations, where catastrophic forgetting is even more threatening and sample efficiency problems are ruled out?

Equations:
3) in eq. 3, the plus should be a minus, as L_BT is a loss term that requires minimization. Also, there shouldn't be a learning rate involved in the look-ahead phase? Do the authors use a learning rate of 1?
4) Eq. 7 should be revised. pi(x) denotes a softmax over the input, without any function being involved. Also, what is the k index in \hat{y}_k referring to?

5) If classification is used to train the slow learner, what is the difference with respect to DER++? There seems to be a significant gap from Tab. 2 (classification-SGD, 67.50) against DER++ in Tab. 1 (62.32). Can the authors provide an intuition for this gap? The two methods seem pretty similar to me in formulation.
6) the authors may consider discussing Meta Experience Replay [2] in their related work, as the REPTILE algorithm [3] employed in such a paper is similar in spirit to the look-ahead optimization used hereby.

Overall, I consider this a borderline submission. I appreciate the idea of using self-supervised learning on the buffer, that seems to bring benefits. However, I have several concerns about experiments, the one about sample efficiency being the most critical.

---
EDIT: after discussions with the authors, I will increase my score from 5 to 7.

References:
[1] Buzzega, Pietro, et al. "Dark experience for general continual learning: a strong, simple baseline." NeurIPS 2020.
[2] Riemer, Matthew, et al. "Learning to learn without forgetting by maximizing transfer and minimizing interference." ICLR 2019.
[3] Nichol, Alex, Joshua Achiam, and John Schulman. "On first-order meta-learning algorithms." arXiv preprint arXiv:1803.02999 (2018).


**Time Spent Reviewing:**

4

---

> ### Author Response · Authors · 2021-08-10
> **Responses to Reviewer RowY**
>
> **Concern#1: Ablation study with only the slow learner.**
>
> Thank you for your suggestion. We conduct an experiment as you suggested where we remove the fast learner and only use the slow learner for both self-supervised and supervised learning.  Please find the results below. We can see that removing the fast learner yields significant performance drops. This result corroborates with our motivation in L35-51 that it is more beneficial to separate the two self-supervised and supervised representations into two distinct systems, which is our proposed DualNet.
>
> | DualNet              | Split miniImageNet-TA |              |              |
> |----------------------|-----------------------|--------------|--------------|
> |                      | ACC                   | FM           | LA           |
> | Slow learner         | 67.36+/-0.62          | 6.40+/-0.37  | 70.73+/-1.03 |
> | Slow + Fast learners | 73.20+/-0.68          | 3.86+/-1.01  | 74.12+/-0.12 |
>
> | DualNet              | Split miniImageNet-TF |              |              |
> |----------------------|-----------------------|--------------|--------------|
> |                      | ACC                   | FM           | LA           |
> | Slow learner         | 27.30+/-0.25          | 34.60+/-1.12 | 59.70+/-1.26 |
> | Slow + Fast learners | 37.56+/-0.52          | 29.46+/-1.68 | 65.00+/-1.79 |
>
> **Concern #2: continual learning protocol and sample efficiency.**
>
> *Concern: “.. optimize only one epoch per task”*
>
> We think online continual learning is more realistic in real-world scenarios; therefore, we directly follow the standard online continual learning setting, which only optimizes each task in one epoch.
>
> *Concern: “methods that are more sample efficient can perform better, but not because of their continual learning capabilities”*
>
> We agree with the Reviewer that the offline continual learning with multiple epochs (equally popular in the literature) would alleviate the sample efficiency problem and allow us to focus more on preventing catastrophic forgetting. However, achieving a good sample efficiency is also an important goal because of a strong prior the model obtained from previous tasks should improve its learning of the newer ones. Therefore, both catastrophic forgetting and forward knowledge transfer (sample efficiency) are bound together, and require continual learning methods to achieve a good trade-off between them [1,2]. As a result, online continual learning over one epoch is a more suitable setting to test the model’s ability to achieve this trade-off, which aligns with our main goal in this study (as discussed in L30-32).
>
> *Concern: “i) the look-ahead classifier, requiring more forward-backward updates for every example (boosting the performances significantly”*
>
> In DualNet, there are two parallel optimization processes: the slow learner’s SSL and the fast learner’s supervised learning. During the supervised learning phase, all methods, including DualNet, performed two updates per incoming sample. The additional updates in DualNet are from the slow learner’s SSL branch, which can perform well even with one update as shown in Table 3, Section 4.4. Moreover, other baselines and DualNet's fast learner are optimized by the standard SGD optimizer; the Look-ahead is only applied to train the slow learner to achieve a better generic representation, which is not directly applicable to other baselines.
>
> *Concern: “DualNet outperforms Offline training.”*
>
> The Offline model uses a standard backbone with no SSL information nor a fast learner. Therefore, it is not a valid upper bound of our DualNet. Existing studies have shown additional information from SSL can improve the performance of the standard supervised-training model without SSL [3,4]. A valid upper bound for DualNet is a model with the same fast and slow learner architectures and optimizes both supervised and SSL objectives in an offline fashion, which we will include in the final version.
>
> *Concern: “ is the number of updates performed by DualNet the same as the one allowed to competitors“*
>
> We follow CTN to cross-validate the number of updates using the validation tasks, which is not encountered again during continual learning. We also note that all models’ performances were saturated with more updates in the supervised learning phase, e.g. 3 and 4. Moreover, as shown in Table 3, even with one slow learner update, DualNet can still outperform other baselines.
>
> *Concern: “results with longer optimization.”*
>
> In the online continual learning setting, the baselines’ performances are saturated after two updates. Therefore, we train DER++ in an offline fashion over 20 epochs per task and compare with DualNet trained in an online fashion. Please find the results below where DualNet's performances are directly taken from Table 1, Section 4.2.
>
> |                   | Split miniImageNet-TA |              |              |
> |-------------------|-----------------------|--------------|--------------|
> |                   | ACC                   | FM           | LA           |
> | DER++ - 20 epochs | 71.96+/-1.36          | 7.26+/-1.50  | 78.23+/-0.85 |
> | DualNet - 1 epoch | 73.20+/-0.68          | 3.86+/-1.01  | 74.12+/-0.12 |
>
> | DualNet           | Split miniImageNet-TF |              |              |
> |-------------------|-----------------------|--------------|--------------|
> |                   | ACC                   | FM           | LA           |
> | DER++ - 20 epochs | 27.50+/-1.83          | 37.52+/-2.83 | 62.30+/-0.43 |
> | DualNet - 1 epoch | 37.56+/-0.52          | 29.46+/-1.68 | 65.00+/-1.79 |
>
>
> The results show that DualNet’s positive results still hold even when DER++ is trained in an offline manner. Moreover, the performance gap remains significant in the challenging task-free setting.
>
> **Concern #3: look-ahead learning rate.**
>
> There are two learning rates in Look-ahead. As provided in appendix B.4, the Look-ahead's SGD learning rate is set to be 3e-4, and the Look-ahead's learning rate ($\beta$ in Eq.4) is set to be 0.05.
>
> **Concern #4: \hat{y}_k meaning.**
>
> \hat{y}_k refers to the snapshot of the model’s logit of the corresponding sample at the end of the k-th task. We will clarify this in the final version.
>
> **Concern #3,4: typos in Eq.3 and in Eq.7.**
>
> Thank you for carefully checking our draft. We will edit these typos accordingly in the final version.
>
> **Concern #5: performance gap between DER++ and classification-SGD DualNet.**
>
> The performance gap (67.50 to 63.48- DER++ with data augmentation) comes from a combination of the increased model capacity from the fast learner and additional experience replay updates. Particularly, the fast learner introduces additional model capacity, which allows the model to optimize for more experience replay steps (during training of the fast and slow learners) and take advantage of the data augmentation to achieve better performance. We also want to note that existing methods may not maximize the model capacity. As shown by our response to Concern #1, Reviewer RowY, where the standard model optimized both the supervised and SSL loss can achieve 67.48 ACC, which is better than other baselines with the same backbone and is close to the classification-SGD DualNet . This result shows that the improvements come from the proposed DualNet framework rather than the simple increased model capacity.
>
> **Concern #6: discussing Meta Experience Replay and Reptile in the related work.**
>
> Thank you for the suggestion. MER uses Look-ahead to improve the sample efficiency in the supervised learning phase. Furthermore, MER only demonstrated promising results on simple datasets such as MNIST. A later study in [5] showed that MER struggles on more challenging benchmarks based on CIFAR, miniImageNet and is outperformed by the standard ER strategy. In contrast, we use look-ahead to improve the SSL of the slow learner, which is more scalable and applicable to challenging benchmarks (as shown in Table 3, 4). We will add this discussion to the final version.
>
>
> [1] Riemer, Matthew, et al. "Learning to Learn without Forgetting by Maximizing Transfer and Minimizing Interference." International Conference on Learning Representations. 2018.
>
> [2] Aljundi, Rahaf, et al. "Online Continual Learning with Maximal Interfered Retrieval." Advances in Neural Information Processing Systems 32 (2019): 11849-11860.
>
> [3] Zoph, Barret, et al. "Rethinking Pre-training and Self-training." Advances in Neural Information Processing Systems 33 (2020).
>
> [4] Newell, Alejandro, and Jia Deng. "How useful is self-supervised pretraining for visual tasks?." Proceedings of the IEEE/CVF Conference on Computer Vision and Pattern Recognition. 2020.
>
> [5]  Chaudhry, Arslan, et al. "On tiny episodic memories in continual learning." arXiv preprint arXiv:1902.10486 (2019).

---

> > ### Comment · Reviewer_RowY · 2021-08-24
> > **Comments**
> >
> > I thank the authors for the detailed response. Here are my comments.
> >
> > **- Ablation study with only the slow learner**
> > I find the results reported by authors remarkable. It still puzzles me a bit how disentangling the two objectives by the proposed masking layers brings such benefits. However, the experiment is solid and I think it could benefit the manuscript.
> >
> > **- Continual learning protocol and sample efficiency**
> > I understand the points made by the authors, but I still have reservations about mixing the problems of catastrophic forgetting and fast learning. I agree that in many online applications we cannot store data for multiple rounds of optimization, and single epochs setups are a way of simulating such situations. I also agree that the ideal continual learning model should be both sample efficient and immune to forgetting. However, at the current state of research, I think it is still early to mix the two and doing so only makes comparisons difficult. Ultimately, I think one pass over the data is indeed the preferred setting, but **provided that a single pass is enough to learn the task**.
> >
> > However, this discussion is not relevant for my judgment on the paper anymore, as the authors provided evidence that DualNet outperforms a DER++ model optimized for many more epochs.
> >
> > **- Offline training is not an upper bound for DualNet**, given the presence of SSL. The authors seem to suggest that SSL, without labels (as in Barlow Twins) can outperform SL with labels. This is not convincing to me as in standard settings (i.e. no transfer learning, no few-shot, etc) SL is considered to be an upper bound for SSL. Thus, I don't understand how using SSL can benefit the slow learner so much it outperforms the joint SL training.
> > EDIT 1: is joint SL training optimized for a single epoch?
> > EDIT 2: what is, in the authors' opinion, a suitable upper bound for DualNet? And why?
> >
> > **- results with longer optimization**. I think this experiment rules out the doubt that DualNet is better than competitors only because of sample efficiency. However, why not optimizing both DER++ and DualNet for 20 epochs? In my opinion, such a comparison would be even better.
> >
> > **- look-ahead learning rate** given that there are two learning rates, shouldn't they both appear in Eq. 3 and 4? Specifically, shouldn't Eq. 3 apply a penalty to the gradient A?
> >
> > RowY

---

> > > ### Author Response · Authors · 2021-08-31
> > > **Response to Reviewer RowY's comment**
> > >
> > > **Comment #1: “how disentangling the two objectives by the proposed masking layers brings such benefits.”**
> > >
> > > We believe disentangling the two objectives brings benefits to continual learning because the SSL representation is less affected by the ever-changing data stream. Therefore, the masking layer can selectively choose which SSL information is related to assist the incoming sample’s SL.
> > >
> > > **Comment #3: DualNet’s upper bound.**
> > >
> > > **EDIT1: is joint SL training optimized for a single epoch?**
> > >
> > > All the offline models are optimized for five epochs.
> > >
> > > **EDIT 2: what is, in the authors' opinion, a suitable upper bound for DualNet? And why?**
> > >
> > > In our work, there are three factors affecting the upper bound model:
> > >
> > > i) model architecture: slow net(standard backbone) vs fast and slow net(our DualNet);
> > >
> > > ii) training loss: supervised learning loss (SL) or supervised and self-supervised learning losses (SL + SSL); and
> > >
> > > iii) data augmentation.
> > >
> > > As a result, we believe that an upper bound of DualNet is a model having all three factors as DualNet (has fast and slow learners, optimized both SL and SSL losses with data augmentation) and is trained offline. The Offline model simultaneously optimizes both the SL and SSL losses, which are backpropagated through both learners.
> > >
> > > To verify this upper bound, we consider the Split miniImageNet benchmark and report the Offline model with different combinations of the above factors in the following table. Note that the configuration **Slow Net + Offline + SL without data augmentation** is the previous model reported in CTN. Our argued upper bound for DualNet has the following configuration: **Fast + Slow nets + Offline + SL + SSL + data augmentation**.
> > >
> > > | Architecture    |    Method    |  Loss  | Data Aug |      ACC     |
> > > |-----------------|:------------:|:------:|:--------:|:------------:|
> > > | Fast + Slow nets | DualNet (CL) | SL+SSL |    Yes   | 73.20+/-0.68 |
> > > |                 |    Offline   |   SL   |    No    | 75.83+/-1.07 |
> > > |                 |    Offline   |   SL   |    Yes   | **77.63+/-0.48** |
> > > |                 |    Offline   | SL+SSL |    Yes   | **77.98+/-0.16** |
> > > | Slow net        |    Offline   |   SL   |    No    | 71.15+/-2.95 |
> > > |                 |    Offline   |   SL   |    Yes   | 75.46+/-0.97 |
> > >
> > > The result confirms the upper bound of DualNet. Moreover, in the offline training with all data, the SSL only contributes a minor improvement to the SL. However, in continual learning, SSL is more beneficial because its representation does not depend on the class label, and therefore more resistant to catastrophic forgetting when old task data is limited.
> > >
> > > **Comment #4: “why not optimizing both DER++ and DualNet for 20 epochs?”**
> > >
> > > We couldn’t report the result of DualNet trained for 20 epochs during the initial rebuttal phase because of the limited time and there were higher priority experiments during that time. We report the results of DualNet training for 20 epochs in the following table.
> > >
> > > |                   | Split miniImageNet-TA |              |              |
> > > |-------------------|-----------------------|--------------|--------------|
> > > |                   | ACC                   | FM           | LA           |
> > > | DER++ - 20 epochs | 71.96+/-1.36          | 7.26+/-1.50  | 78.23+/-0.85 |
> > > | DualNet - 1 epoch | 73.20+/-0.68          | 3.86+/-1.01  | 74.12+/-0.12 |
> > > | DualNet - 20 epochs| 75.20+/-0.09   | 6.08+/-0.23 | 78.99+/-0.10 |
> > >
> > > | DualNet           | Split miniImageNet-TF |              |              |
> > > |-------------------|-----------------------|--------------|--------------|
> > > |                   | ACC                   | FM           | LA           |
> > > | DER++ - 20 epochs | 27.50+/-1.83          | 37.52+/-2.83 | 62.30+/-0.43 |
> > > | DualNet - 1 epoch | 37.56+/-0.52          | 29.46+/-1.68 | 65.00+/-1.79 |
> > > | DualNet - 20 epochs | 38.26+/-1.10 | 31.83+/-0.93 | 66.63+/-0.61 |
> > >
> > > Similar to DER++, we observe an improvement in the learning of each task (LA), which is more significant than the increases in catastrophic forgetting and results in better final performance (ACC).
> > >
> > > **Comment #5: “shouldn't Eq. 3 apply a penalty to the gradient A?”**
> > >
> > > We apologize for the confusion caused by Eq.3, where we absorbed the learning rate into the optimizer $\mathcal{A}$. To avoid future confusions, we will amend the Eq. 3 and 4 as:
> > > $$\mathrm{Eq.3:} \quad \tilde{\phi}\_k \gets \tilde{\phi}\_{k-1} - \epsilon \nabla\_{\tilde{\phi}\_{k-1}} \mathcal{L}_{\mathcal{B} \mathcal{T}}, \quad \mathrm{with} \quad \tilde{\phi}_0 = \phi \quad \mathrm{and} \quad k=1,2,\ldots,K $$
> > > $$\mathrm{Eq.4:} \quad \phi \gets \phi + \beta (\tilde{\phi}_K - \phi)$$
> > > Where $\epsilon$ is the Look-ahead’s SGD learning rate and $\beta$ is the Look-ahead learning rate.
> > > Our experiments set the learning rates as: $\epsilon=3e-4$ and $\beta=0.05$.
> > >
> > > Please let us know if you still have any concerns and we will be happy to address them.

---

> > > > ### Comment · Reviewer_RowY · 2021-09-01
> > > > **Response**
> > > >
> > > > Thank you for your further response. I have no further doubts and I will increase my score to acceptance.
> > > >
> > > > Best,
> > > > RowY

---

> > > > > ### Author Response · Authors · 2021-09-01
> > > > > **Response**
> > > > >
> > > > > We are delighted that our responses addressed your concerns. We thank the Reviewer for carefully assessed our work and changed the score accordingly after the discussion.
> > > > >
> > > > > Best.

---

### Official Review · Reviewer_xfxT · 2021-07-19

**Rating:** 6
**Confidence:** 4

**Summary:**

Inspired by Complementary Learning Systems (CLS), the paper proposes a framework for performing continual learning which is composed of self-supervised learning stream (aka slow learner) that conducts representation learning and of supervised learning stream (fast learner) that attains knowledge from labeled samples via adapting slow learner's representation. The DualNet outperforms other methods on CORE50 and miniImageNet.

**Ethical Concerns:**

Yes.

**Ethics Review Area:**

["I don’t know"]

**Limitations And Societal Impact:**

Yes.

**Main Review:**

(+) The paper considers both multi-head and single-head (no task identifier) evaluations.

(+) The paper offers a concrete philosophy for continual learning. DualNet is based on CLS which is what CTN was based on.

(-) There are many criterion for choosing the appropriate SSL. It would be nice to have performance comparison between the different SSL and discussion as to why one would perform better than the other. Table 2 compares only to SimCLR but not SimCLR v2, BYOL, MoCo.

(-) The proposed network is evaluated on only two dataset whereas many continual learning papers evaluate their algorithm on at least 3 (Permuted MNIST Alternating CIFAR 10 and CIFAR 100, Multiple datasets learning). The proposed network should be validate for much wider set of datasets.

(-) There are many reference that are missing in the paper. Sebastian Lee · Sebastian Goldt · Andrew Saxe, Continual Learning in the Teacher-Student Setup: Impact of Task Similarity ICML 2021 Abhishek Kumar · Sunabha Chatterjee · Piyush Rai Bayesian Structural Adaptation for Continual Learning ICML 2021

**Time Spent Reviewing:**

3

---

> ### Author Response · Authors · 2021-08-10
> **Responses to Reviewer xfxT**
>
> **Concern #1: comparison with more SSL methods.**
>
> Thank you for the suggestion. Our main hypothesis  of this work is to validate if  SSL representations are beneficial to continual learning in addition to supervised learning. As noted in L146-148, our framework is compatible with any existing SSL methods and we expect that a better SSL method can be directly transferred to a better performance in DualNet. We verified this in Section 4.2, Table 2 (Look-ahead), where the Barlow Twin loss achieved better performance than SimCLR, and both are better than the standard supervised training of the slow learner without SSL.
> We chose these two methods because they do not add additional memory overhead compared to other SSL methods such as MOCO, which we discussed in L110-112.
> We will try to include more SSL methods in the final version, but exhaustively comparing all SSL methods is beyond the main focus of this study.
>
> **Concern #2: evaluation on more datasets.**
>
> We didn’t include experiments on the MNIST and CIFAR datasets mainly because these datasets are much easier and the performances on such datasets are quite saturated, e.g. DER[1] and CTN[2] performances are already close to the Offline training model on MNIST-based benchmarks. In contrast, the Split miniImageNet and CORE50 benchmarks we considered are more challenging and carefully  designed for recent continual learning studies. Please find a comparison with ER and CTN[2], a state-of-the-art method, on the Split-CIFAR100 TA benchmark in the table below. We will try to add this benchmark to the final version.
>
> | Method  | Split-CIFAR100 TA |             |              |
> |---------|-------------------|-------------|--------------|
> |         | ACC               | FM          | LA           |
> | ER      | 61.36+/-1.01      | 7.20+/-0.72 | 67.05+/-1.08 |
> | CTN     | 67.65+/-0.43      | 6.33+/-0.70 | 73.43+/-0.45 |
> | DualNet | 71.70+/-0.46      | 4.92+/-0.34 | 74.16+/-0.71 |
>
>
> **Concern #3: missing references.**
>
> Thank you for the references. However, both papers were accepted at ICML 2021 with the camera ready date after the NeurIPS submission deadline, making it impossible for us to compare with them during the initial submission. Moreover, both papers explore orthogonal research directions to ours such as theoretical study to task similarity or the Bayesian approach to continual learning. Our submission already covered references up to ICLR2021 and we will be happy to discuss newer ICML2021 papers in the final version.
>
> [1] Pietro Buzzega, Matteo Boschini, Angelo Porrello, Davide Abati, and Simone Calderara. Dark experience for general continual learning: a strong, simple baseline. In 34th Conference on Neural Information Processing Systems (NeurIPS 2020), 2020.
>
> [2] Pham, Quang, Chenghao Liu, and Doyen Sahoo. "Contextual transformation networks for online continual learning." International Conference on Learning Representations. 2020.

---

> ### Author Response · Authors · 2021-09-03
> **Additional Experiments**
>
> We have conducted experiments regarding additional SSL losses to respond to your concerns. Please find the result in the below tables.
>
> **Update 1 (Sep 10th): adding the Split CIFAR10/100 results.**
>
> **Update 2 (Sep 15th): adding the BYOL results.**
>
> **Concern #1: “It would be nice to have performance comparison between the different SSL”**
>
> Please find the result of the DualNet with SimSiam[1] and BYOL[2] as the objective to train the slow learner in the following table.
>
> | Objective      |              |     SGD     |              |              |  Look-ahead |              |
> |----------------|--------------|:-----------:|--------------|--------------|:-----------:|--------------|
> |                |      ACC     |      FM     |      LA      |      ACC     |      FM     |      LA      |
> | BarlowTwins    | 64.20+/-2.37 | 4.79+/-1.19 | 64.83+/-1.67 | **73.20+/-0.68** | **3.86+/-1.01** | **74.12+/-0.12** |
> | SimCLR         | **71.49+/-1.01** | **4.23+/-0.46** | **72.64+/-1.20** | 72.13+/-0.44 | 4.13+/-0.52 | 73.09+/-0.16 |
> | SimSiam        | 70.55+/-0.98 | 4.93+/-1.31 | 71.90+/-0.65 | 71.94+/-0.64 | 4.21+/-0.28 | 72.93+/-0.38 |
> | BYOL       | 69.76+/-2.12 | **4.23+/-1.41** | 70.33+/-0.87 | 71.73+/-0.47 | **3.96+/-0.62** | 72.06+/-0.28 |
> | Classification | 67.50+/-1.07 | 7.86+/-0.80 | 71.93+/-0.89 | 70.96+/-1.08 | 6.33+/-0.28 | 73.92+/-1.14 |
>
> Overall, we still have similar conclusions with our existing experiments: (i) SSL losses are generally better than the classification loss to train the slow learner; and (ii) the Look-ahead optimizer is more suitable to train the slow learner.
>
> **Concern #2: additional benchmarks**
>
> In addition to the Split CIFAR100-TA benchmark reported in our initial response, we further conducted the **Split CIFAR10/100-TA** benchmark and report the results in the following table. The Split CIFAR10/100 benchmark is constructed by using the CIFAR10 as the first task, followed by 10 additional tasks constructed by splitting the CIFAR100 datasets in to a sequence of 10 tasks. The reported results follow the online, task-aware setting.
>
> | Method  | Split CIFAR10/100 TA |             |              |
> |---------|-------------------|-------------|--------------|
> |         | ACC               | FM          | LA           |
> | ER      | 61.20+/-1.13      | 8.50+/-1.02 | 68.80+/-0.34 |
> | CTN     | 66.07+/-0.21      | 5.43+/-0.99 | 70.96+/-0.62 |
> | DualNet | 68.03+/-0.52      | 5.42+/-0.42 | 72.90+/-0.49 |
>
> Consistent with the results on the Split miniIMN and CORE50 benchmarks reported in the main paper, we still observe positive results of DualNet on the Split CIFAR100 and Split CIFAR10/100 benchmarks conducted during the discussion period.
>
> If our responses have addressed your concerns, could you please change the score to reflect it? Otherwise, please let us know if you still have any remaining questions and we will be happy to address them.
>
> [1] Chen, Xinlei, and Kaiming He. "Exploring simple siamese representation learning." Proceedings of the IEEE/CVF Conference on Computer Vision and Pattern Recognition. 2021.
>
> [2] Grill, Jean-Bastien, et al. "Bootstrap Your Own Latent: A new approach to self-supervised learning." Neural Information Processing Systems. 2020.

---

> > ### Author Response · Authors · 2021-09-15
> > **Authors Response Summary**
> >
> > We thank the Reviewer xfxT for the comments. We have clarified the criterion for choosing the SSL loss (concern #1) and additional references to the ICML 2021 papers (concern #3) in our initial response. In addition, we conducted additional experiments as suggested by the Reviewer and reported in our follow up responses. Particularly, we provided the results of the Split CIFAR100 and Split CIFAR10/100 benchmarks (concern #2). We also implemented two additional SSL losses (SimSiam and BYOL) to our DualNet framework (concern #1). Across all the new experiments, our DualNet still showed positive results compared to other baselines and the benefit of SSL remained consistent with our discussion.
> >
> > We hope our discussions and additional experiment results have addressed your concerns and you could adjust your rating accordingly. If you still have any questions that require further clarifications, please let us know so that we could address them.
> >
> > Thank you,
> >
> > Authors

---

> > > ### Comment · Reviewer_xfxT · 2021-09-25
> > > **After reading rebuttal and comments of other reviewers, I change my score to accept.**
> > >
> > > I thank the authors for the rebuttal. If the authors can incorporate the additional experimental results and discussions in the final revised manuscript, the credibility of the paper will be much improved.
> > > I change my rating to accept.

---

> ### Comment · Reviewer_xfxT · 2021-09-25
> **After reading the rebuttal and reviews of other reviewers, I change my final rating to accept.**
>
> I believe that with the additional experiments and discussion, the credibility of the paper is considerably improved. I urge the authors to incorporate additional experiments and discussion into the revised manuscript.

---

### Author Response · Authors · 2021-08-10
**General Reply to All Reviewers**

We thank the Reviewers for insightful comments and valuable feedback. We are delighted that they found our work to be novel (R RowY, qVwg) and well-motivated (R xfxT). They (R xfxT, RowY, qVwq) agree that the paper is well-written, is based on concrete philosophy, and achieved encouraging results.
There is a shared concern between R RowY and R qVwg regarding the sample efficiency and a fair comparison with the baselines. We clarify that all the methods are compared in the same settings by following the standard online continual learning setup in literature. To explicitly answer the concern of sample efficiency, we further provide an additional experiment where we rule out the sample efficiency problem by allowing a baseline to train for more epochs. The detailed discussion and experimental results are provided in our response to R RowY, concerns #2.

We appreciate the Reviewers’ comments and will incorporate all feedback and additional experiments in the final version.

---

### Decision · Program_Chairs · 2021-09-27

**Decision:**

Accept (Poster)

**Comment:**

This paper provides a dual-learning mechanism that uses a novel combination of a separated representation learner with a lean feature adaptation model.  The reviewers found their approach convincing and that it performs well against current competitors. They also found this paper to be well written, and after discussion on various technical aspects, quite compelling.  The one slightly negative review provided little justification for their points and chose not to engage with the authors after the authors provided a detailed rebuttal against the points raised in the more negative review.